# The Utilization of National Tobacco Cessation Services among Female Smokers and the Need for a Gender-Responsive Approach

**DOI:** 10.3390/ijerph18105313

**Published:** 2021-05-17

**Authors:** Ahnna Lee, Kang-Sook Lee, Dahyeon Lee, Hyeju Ahn, Hyun-Kyung Lee, Hyekyeong Kim, Jakyoung Lee, Hong-Gwan Seo

**Affiliations:** 1Department of Public Health, Graduate School, The Catholic University of Korea, Seoul 06591, Korea; abcde@catholic.ac.kr; 2Department of Preventive Medicine, College of Medicine, The Catholic University of Korea, Seoul 06591, Korea; dada2020@catholic.ac.kr; 3Department of Health Promotion, Graduate School of Public Health, The Catholic University of Korea, Seoul 06591, Korea; 4Korean Association on Smoking or Health, Seoul 07238, Korea; heyjude@kash.or.kr (H.A.); lovhyun@kash.or.kr (H.-K.L.); hkkim@ewha.ac.kr (H.K.); jakyl00722@gmail.com (J.L.); 5Department of Health Convergence, Ewha Womans University, Seoul 03760, Korea; 6Graduate School of Public Health, Yonsei University, Seoul 03722, Korea; 7Department of Family Medicine, National Cancer Center, Gyeonggi-do 10408, Korea

**Keywords:** smoking cessation, female smoker, national tobacco cessation service, gender-responsive tobacco control

## Abstract

Despite the steadily increasing prevalence of female smoking, gender-responsive tobacco cessation services have not been widely provided worldwide. The purpose of this study is to identify factors associated with the use of tobacco cessation services among female tobacco product users in Korea from a national perspective. We performed a logistic regression analysis using data from 663 female smokers; 11.0% of female smokers had used government-supported smoking cessation services. A logistic regression model showed a statistically significant association between the utilization of smoking cessation services and a history of pregnancy and childbirth, depression, current use of heated tobacco products and multiple tobacco products, parental smoking status and receiving advice to quit. With regard to the motivation ruler, those in their 50s reported a higher importance than those in their 20s. Weight gain concerns when quitting smoking were the lowest among the participants aged 19–29. The need to develop gender-specific smoking cessation programs is the highest among the participants aged 39–49 and the lowest among those aged 19–29. This study suggests several factors related to the utilization of national health services among female smokers. Further studies considering gender-specific needs for the development of gender-responsive tobacco cessation support are needed.

## 1. Introduction

Worldwide, there have been more countries that have achieved a significantly greater decrease in smoking prevalence among men than women. Although the prevalence of male tobacco use still remains higher than that of females globally, there has only been a minimal decline in female smoking and even an increase in many low- and middle-income countries [1]. In Korea, while the smoking rate has dropped from 66.3% to 36.7% in adult males in the past 20 years, the adult female smoking rate has increased from 6.5% in 1998 to 7.5% in 2018. In particular, the prevalence of smoking among women in their 20s–40s has more than doubled from 4.4% to 10.9% in the past 20 years in Korea [2]. Considering the cultural background of social disapproval of female smoking in Asian countries, the actual smoking rate among Korean women may be underestimated due to the under-reporting of smoking status [3,4].

One of the key strategies to reduce demand for tobacco provided by the World Health Organization (WHO) Framework Convention on Tobacco Control (FCTC) is contained in Article 14: the demand reduction measures concerning tobacco dependence and cessation [5]. In 2019, the WHO discussed “Offer help to quit tobacco use” in their periodic reports on the global tobacco epidemic. As of 2018, 32% of the global population including Koreans was covered by comprehensive national tobacco cessation support. This best-practice adoption of cessation included three elements: (1) cost coverage of cessation advice in clinical or community settings, (2) cost coverage of nicotine replacement therapy and (3) national toll-free quit lines [6].

Smoking cessation and smoking cessation treatments are regarded as a cost-effective way to reduce smoking-induced morbidity and mortality [7]. Combining behavioral support for tobacco cessation from an expert and pharmacotherapy increases the likelihood of successful cessation compared with using no cessation support [8]. Previous research has addressed factors associated with the utilization of smoking cessation services including insurance status, receiving advice from healthcare professionals [9], a diagnosis of mental health and the self-efficacy to quit [10].

Studies have shown gender differences in smoking cessation and that women find it more difficult to quit tobacco use than men [11]. Compared with men, women show a lower utilization of smoking cessation services [12]. Due to stigma-related barriers, female smokers tend to be reluctant to reveal their smoking status, resulting in less usage of cessation support. Regarding user characteristics of national smoking cessation services in Korea, female smokers reported more usage of web- and telephone-based cessation services, which can provide more privacy than public health center-based smoking cessation clinics [13].

To prevent an increase in female smoking prevalence, gender-responsive measures for tobacco control are necessary particularly in the changing social context regarding gender. The FCTC recommends incorporating gender into all measures for tobacco control, to increase access to health services for tobacco dependence among marginalized populations including women and to train healthcare providers to consider sex and gender specificities [14]. However, there is insufficient evidence that tobacco control has been sufficiently gender-responsive. Calls have been made to systematically integrate the analysis of sex and gender in program planning [15].

Thus, the aim of this study is to examine factors associated with the utilization of national smoking cessation services and to explore the gender-specific characteristics of service users to develop effective recruitment strategies for national cessation services.

## 2. Materials and Methods

### 2.1. National Tobacco Cessation Services

In accordance with its guidelines for the implementation of FCTC Article 14 [5], quitting support strategies and tobacco treatment infrastructure, the Korean government launched the first nationwide public health center-based tobacco cessation services in 2005 [16]. In 2006, the government started to provide a toll-free quit line offering telephone counseling. Strengthened cessation support was initiated in 2015 including health insurance covering tobacco dependence treatment and outreach services for socially marginalized smokers such as women (Figure 1). These comprehensive nationwide tobacco cessation services enabled a significant decrease in smoking prevalence in men from 66.3% in 1998 to 36.7% in 2018 [2].

### 2.2. Data Collection

From an online panel of a professional research company, namely, the Hyundae Research Institute, 708 Korean female smokers over the age of 19 were sampled. A proportional quota sampling method was used when recruiting the participants to ensure that the respondents demographically represented the general population with quotas based on female smoking rate by age group. Reference data were derived from the 2018 Korea National Health and Nutrition Examination Survey, which was based on the Korean population and housing census. Emails and text messages with a link to a survey questionnaire were sent to random panel members. The research panel included both males and females of all age groups and smokers and those who had never smoked. A nationwide web survey was conducted from 4–9 December 2020. At the beginning of the web survey, Yes or No questions asking the respondents whether they had used any tobacco product in the past five years, whether their age was over 19 and whether they were female were added. With the disqualification logic, the survey only included adult female respondents who had used any tobacco product at least once within the past five years. We used a self-designed questionnaire based on a previously validated nationwide health and tobacco-related survey. Informed consent was obtained from each participant during the survey.

### 2.3. Measures

#### 2.3.1. The Utilization of National Tobacco Cessation Services

National tobacco cessation services provided by the Ministry of Health and Welfare include smoking cessation clinics at 256 public health centers, national Quitline services, residential five-day programs (regional smoking cessation centers), smoking cessation outreach services (regional smoking cessation centers) and health insurance-covered smoking cessation treatment services. The participants who selected any of these services to the question “Which smoking cessation service have you experienced?” were defined as users of national smoking cessation services. 

#### 2.3.2. General Characteristics

Demographic variables such as age, occupation, marital status, living arrangement and history of pregnancy and childbirth were included. We categorized occupations as (1) non-manual workers (general managers, professionals and office workers), (2) sales/service workers, (3) manual workers (agriculture, fishery and forestry workers, machine operation and assembly workers and craft and related trade workers) and (4) others (students, homemakers and unemployed). Marital status was categorized as (1) single and (2) married (including separated/divorced/widowed). Living arrangements were measured by asking whom they were living with and the response options were partners, children (son/daughter), grandchildren, parents, brothers/sisters, son/daughter-in-law, grandparents, relatives, friends and living alone. We then classified living arrangements into two groups: living alone for those who answered “living alone” and those who did not live alone for the rest of the response options. History of pregnancy and childbirth included former experience of (1) pregnancy and childbirth, (2) pregnancy only and (3) no history. 

#### 2.3.3. Mental Health

Depression was assessed using the following question: “Have you ever had feelings of sadness or hopelessness every day for two weeks during the last twelve months?”. The response options were yes and no. Perceived stress was measured using the following question: “How would you rate your level of usual stress?”. The responses included five options, namely, “very high”, “high”, “moderate”, “low” and “none” and were regrouped into three categories: “very high/high” as “high”, “moderate” and “low/none” as “low”.

#### 2.3.4. Tobacco-Related Characteristics

In this study, smokers were defined as individuals who had used any tobacco product once in their life. The survey only included respondents who had used a tobacco product in the past five years among smokers. Tobacco-related characteristics included the age of smoking initiation, current use of tobacco products, current status of poly tobacco users, smoking status of people around the participants and receipt of advice to quit smoking. Current smokers were defined as individuals who had used a tobacco product in the past 30 days. The respondents were asked to report the tobacco products they currently used. We then classified the respondents as a current cigarette, electronic cigarettes (e-cigarettes) and heated tobacco product (HTP) user from answers if they reported any current use of cigarettes, e-cigarettes and HTPs. Additionally, the respondents were classified as current multiple tobacco product users if they reported the current use of two or more tobacco products. The smoking status of people around the participants was measured by asking whether their parents, partners and friends were smokers and the response options included yes and no. The participants were asked whether they had someone around them giving them advice to quit smoking. The response options included parents, brothers/sisters, spouse/lover, children/grandchildren, friends/upper and lower classmates, colleagues, teachers, medical personnel, others and none. The responses were regrouped into two categories: “none” as “no” and the rest as “yes”. 

#### 2.3.5. Motivation Ruler for Tobacco Cessation

The motivation ruler [17] was assessed by three 10-point rulers related to the motivation to quit smoking: (1) the importance ruler: “How important is smoking cessation to you?” (0 = not at all important, 10 = 100% important); (2) the readiness ruler: “How ready are you to quit smoking?” (0 = not at all ready, 10 = 100% ready) and (3) the confidence ruler: “How confident are you that you will quit smoking? (0 = not at all confident, 10 = 100% confident)”.

#### 2.3.6. Concerns about Weight Gain

Post-cessation weight gain concerns were measured using a 10-point ruler as a visual aid: “Are you worried about gaining weight as you quit smoking? (0 = not at all worried, 10 = 100% worried)”.

#### 2.3.7. The Need for Gender-Specific Smoking Cessation Services

Regarding the need for gender-specific smoking cessation services, the survey participants were asked the following questions by using a five-point Likert scale: “Do you think that smoking cessation counselors should provide smoking cessation support that takes into account female characteristics during individual counseling? (1 = strongly disagree, 5 = strongly agree).

### 2.4. Data Imputation

Of the total 708 sampled participants, one participant with an inconsistent response was excluded. Of the 707 participants, the subjects with missing values were replaced using a multiple imputation procedure with the assumption that data are missing at random [18]. A simulation technique such as Markov Chain Monte Carlo, which generates random draws from non-standard distributions via Markov chains, was used for multiple imputation [19]. The following variables of missing data were included: the utilization of national tobacco cessation services, current cigarette use, current e-cigarette use, current HTP use, current multiple tobacco product use, intention to quit, three motivation rulers (importance, readiness, confidence) and needs for gender-specific smoking cessation services. Of the 707 participants, 44 participants who were not aware of national tobacco cessation services were excluded from the analysis. A total of 663 female smokers were included for the final sample in the analysis.

### 2.5. Statistical Analysis

We first conducted a descriptive analysis to examine the general and smoking-related characteristics of the survey participants. A Pearson’s chi-squared test and *t*-tests were used to test the statistical significance of the differences in the use of smoking cessation services. In this step, the properties that were to be included in the multiple regression model were determined. Parameters with a *p*-value below 0.05 were included in the logistic regression model. We then performed simple and multiple logistic regression analyses to identify statistically significant factors of service utilization with an adjustment for confounding variables. The second model added two interaction terms: one between current HTP use and age and one between current multiple tobacco product use and age. A multinomial regression was used to evaluate the factors that affected the number of utilized service types. The results from the logistic regression analysis were presented as odds ratios (ORs) with 95% confidence intervals (95% CI). The motivation rulers for tobacco cessation, weight gain concerns and the need for a gender-specific smoking cessation program among different age groups were evaluated with a one-way analysis of variance with a post-hoc Scheffé’s test for multiple comparisons. Regarding the normality of all of the variables in the analysis, none of the variables were highly skewed or kurtotic. The absolute skewness and kurtosis of the variables larger than 2 and 7 were considered non-normal, respectively [20]. An alpha level of 0.05 was considered as the cutoff for significance. R statistical software version 4.0.4(R Foundation for Statistical Computing, Vienna, Austria) and SPSS version 27.0 (SPSS, Inc., Chicago, IL, USA) were used for all analyses.

## 3. Results

### 3.1. Demographic and Smoking-Related Characteristics of the Survey Participants

The basic characteristics of the study participants are shown in Table 1. In this study, 89.0% (590 of 663) of the respondents had never used national smoking cessation services and 11.0% (73 of 663) of the respondents had used one. Regarding marital status among smokers, users of smoking cessation services were more likely to be married than single (*p* = 0.002). Moreover, those who had a history of both pregnancy and childbirth reported a significantly greater use of smoking cessation services (69.9% vs. 46.4%, *p* < 0.001). Compared with those without the experience of government-supported smoking cessation services, those with experience expressed more depressive moods. We found that the current use of heated tobacco products (HTPs) and multiple tobacco products were more prevalent among national service users (*p* = 0.003 and *p* = 0.001). In addition, users of smoking cessation services were more likely to have smoking parent(s) than those who had never used. With regard to smoking cessation advice, service users received more quitting advice than those who had never used (93.2% vs. 83.4%, *p* = 0.030).

### 3.2. Logistic Regression Analysis of Factors Associated with the Utilization of National Tobacco Cessation Services

A simple logistic regression analysis revealed a statistically significant association between those who had ever used national smoking cessation services and a history of pregnancy and childbirth (OR = 2.60, 95% CI: 1.51–4.47). Those who reported depressive moods had more than twice the odds of using smoking cessation services (OR = 1.94, 95% CI: 1.18–3.20). Smokers currently using HTPs had an OR of 2.10 (95% CI: 1.29–3.42) for using national smoking cessation services. Current multiple tobacco product use was associated with twice higher odds of using national smoking cessation services (OR = 2.21, 95% CI: 1.35–3.63). Compared with smokers with non-smoking parents, the odds of using national smoking cessation services increased among smokers with smoking parents (OR = 1.82, 95% CI: 1.10–3.01). Among smokers who had received advice to quit smoking, the likelihood of using government-supported smoking cessation services was increased (OR = 2.71, 95% CI: 1.06–6.89). Age, marital status, history of pregnancy and childbirth, depression, current HTP use, multiple tobacco product use, parent(s) smoking status and the receipt of advice to quit were included in Model 1 as confounding factors. After adjusting for confounding variables, a statistically significant association between the utilization of smoking cessation services and depression and parental smoking status was revealed. In Model 2, there were no significant interactions between current HTP use and age and current multiple tobacco product use and age (Table 2).

### 3.3. Multinomial Logistic Regression Analysis of Factors Associated with the Number of Utilized Service Type

In Table 3, a multinomial logistic regression analysis revealed that reporting depressive moods was associated with twice higher odds of using a single type of tobacco cessation support (OR = 2.06, 95% CI: 1.15–3.69). Among smokers whose partner was a smoker, the likelihood of using more than two types of national tobacco cessation service was increased (OR = 3.24, 95% CI: 1.06–9.87).

### 3.4. The Survey Participants’ Motivation Ruler for Smoking Cessation and Concern about Weight Gain

In terms of motivation rulers for tobacco cessation, users of smoking cessation services provided by the government tended to report significantly higher importance than those who had never used. Post-hoc tests revealed that those in their 50s displayed a higher importance than those in their 20s (*p* = 0.003, Scheffé’s test). Users of smoking cessation services reported more weight gain concerns when quitting smoking than those who had never used. Those in their 20s indicated the lowest weight gain concerns when quitting smoking among all age groups (Table 4).

### 3.5. The Survey Participants’ Need for Gender-Specific Smoking Cessation Services

When asked about the need for gender-specific services for smoking cessation, users of government-supported smoking cessation programs reported higher mean scores than those who had never used. Among the different age groups, those in their 20s and 40s reported the lowest and highest mean scores, respectively (Table 5). 

## 4. Discussion

To our knowledge, this nationwide study is the first to report the factors associated with the use of government-supported smoking cessation services among female smokers in Korea. We observed a few characteristics associated with the utilization of national smoking cessation services. A few of these factors included general characteristics (history of pregnancy and childbirth, depression) and tobacco-related characteristics (current use of HTPs, current use of multiple tobacco products, parental smoking status, receiving advice to quit smoking). This study showed the differences regarding weight gain concerns among users and those who had never used tobacco cessation support. Our study also provided female smokers’ opinions regarding the need for the implementation of gender-specific and gender-sensitive interventions.

This study showed that service users of national tobacco cessation support were more likely to have a history of pregnancy and childbirth. Studies [21,22] have shown that pregnancy and parenting provide a powerful motivation for many female smokers to quit smoking. Mothers of young children tend to be more responsive to interventions for smoking cessation [23]. In a qualitative study [24] conducted among Korean female smokers, younger smokers reported a rather opposed attitude to current smoking cessation services to emphasize the harmful effects of smoking related to pregnancy and childbirth. They recognized that women seem to be considered only within the context of reproductive health; only for giving birth, rather than as individuals equal to men. When providing a gender-responsive tobacco cessation program, whether current pregnancy- and childbearing-focused programs could be suitable for all age groups should be considered.

Our findings that those with depressive symptoms were associated with increased odds of using national tobacco cessation support may be explained in line with previous studies. A four-country International Tobacco Control study [25] observed that female smokers with depressive symptoms made more attempts to quit but experienced more smoking relapses. A depressive mood has been identified as part of nicotine withdrawal [26]. Nicotine in tobacco products binds to α4β2 nicotinic acetylcholine receptors in the brain and triggers the release of the neurotransmitter dopamine, which helps to create a good feeling [27]. Once a smoker stops smoking and the brain lacks dopamine, a low mood may be experienced [28]. Moreover, as studies have found sex-specific differences in cravings to smoke when experiencing negative mood situations [29], attention to psychological and emotional factors when counseling women seems to be important. To provide effective gender-responsive tobacco cessation support, different stimulations and concerns regarding smoking need to be considered in both genders.

In our analysis, the participants who used the national tobacco cessation support were associated with the current use of HTPs. A possible explanation for the greater odds of using cessation support in current HTP users may be that these products are considered smoking cessation aids [4]. A recent study [30] among Korean adolescents revealed that HTP use among current combustible cigarette smokers was associated with higher odds of having attempted to quit compared with exclusive cigarette users. Additionally, current multiple product users were more likely to report the use of the national cessation support in this study. Previous studies among Korean [31] and US [32] adults found that past quitting attempts were associated with poly-tobacco use. Given that a considerable proportion of HTP users are dual or multiple users [33], efforts should be made for this motivation towards smoking cessation to be connected to successful complete cessation. 

In this study, users of national smoking cessation services were more likely to have smoking parents. Parental smoking has been found to be a predictor of smoking in children in previous literature [34]. Adolescents who have smoking parents are more likely to initiate smoking at an earlier age [35], which can lead to problematic smoking trajectories [36] requiring smoking cessation support. There is another possibility that parents who struggle to quit smoking may have negative attitudes toward tobacco smoking, which can be relayed to their children [37]. In addition, as our study population was over the age of 19, the role of parental smoking in adult children may be different from that in adolescence. Therefore, a further analysis is needed regarding the relationship between parental smoking status and adult children’s usage of smoking cessation support. 

Quitting advice has been identified as an important determinant of the utilization of smoking cessation services. Our study found that smokers who received quitting advice from people in close relationships or healthcare professionals showed an increased use of smoking cessation services. Quitting advice from a healthcare professional has been reported to promote smoking cessation attempts and increase the possibility of successful cessation [38]. In a study [39] on spouses’ support related to tobacco cessation, smokers who maintained successful abstinence for one, three and six months reported a higher frequency and perceived helpfulness from partner interactions. Data from the US National Adult Tobacco Survey demonstrated that healthcare professional advice was correlated with a higher utilization of a Quitline [40]. 

Documented studies [41,42] have shown that many female smokers have fears related to post-cessation weight gain. Our data showed that smokers with concerns about gaining weight requested more smoking cessation support. A meta-analysis [43] of 62 studies revealed that an average 4–5 kg increase in body weight after 12 months of abstinence was associated with tobacco cessation. Smokers with a higher nicotine dependence are more likely to gain weight [44]. In addition, compared with younger women, older women are known to gain more weight when quitting tobacco products [45]. As weight gain concerns varied among different age groups in this study, a customized and targeted approach would be helpful when counseling for weight management in female smokers. 

The participants in this study were asked to evaluate their agreement on the need for individualized gender-specific tobacco cessation support in counseling. Smokers with experience with a national smoking cessation service expressed more agreement on the need for a gender-specific approach in this study. Evidence [46] suggests that the needs of female smokers for smoking cessation programs include choosing from a diverse range of services and female-specific educational topics. In our data, the participants in their 20s were less in favor of female-focused smoking cessation services than those in their 40s. In a qualitative study [24], smokers in their 20s expressed resentful concerns about current pregnancy- and childbirth-focused approaches for tobacco cessation in female smokers. Tobacco cessation support developed for women had put specific emphasis on smoking during pregnancy. Although this is crucial for both the woman and the fetus, it became necessary to develop broader programs to support tobacco cessation for women throughout their lives [47]. Tailored smoking cessation interventions that consider individual challenges in a life cycle can be beneficial. Future research is warranted regarding the perception of specific gender-responsive approaches in different age groups. 

### Limitations

Our study has several limitations. First, the sample of the present study was obtained from a research panel, thus limiting its generalizability. While efforts were made to ensure that the survey sample represented the population with a proportional quota sampling design, the sample may not be representative of the overall population. Thus, the statistical inference in this study was for the sample not the general population. Given the findings of this study, further research with a nationally representative sample is needed to explore gendered factors associated with the usage of tobacco cessation support. Second, the relatively high prevalence of usage of electronic nicotine delivery systems and HTP can be explained by the inclusion criteria in this study. As this study only included panels who had used a tobacco product in the past five years in order to reflect on newly developed national tobacco cessation services, the usage rates of the new and emerging tobacco products might be higher than that of general population. In addition, the online survey method may be related to a higher prevalence of female smoking than can be found in general by ensuring respondents’ anonymity. The study results therefore need to be interpreted with caution. Finally, regarding the nature of the observational study design, recall bias could occur when the participants were asked about their past experiences. Future research with varied study designs will be helpful to understand the characteristics of female tobacco product users with the intention to quit. Despite the study’s limitations, using nationwide survey data among female smokers with a quantitative research design to examine the use of national tobacco cessation services was a notable strength.

## 5. Conclusions

This study suggested several factors associated with the use of national health services. The results could be further used to examine what promoted and hindered the use of national tobacco cessation services among female smokers. Future studies on these related factors will be crucial to improve our understanding of the specific needs for developing gender-sensitive tobacco control policies.

## Figures and Tables

**Figure 1 ijerph-18-05313-f001:**
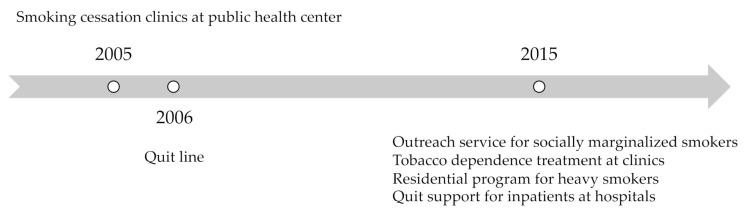
Introduction of national tobacco cessation support in the Republic of Korea.

**Table 1 ijerph-18-05313-t001:** Demographic and smoking-related characteristics of the survey participants (*n* = 663).

Variables	National Tobacco Cessation Services			*p* Value ^†^
	Total n(%)	Ever Used n(%)	Never Used n(%)	
	663 (100)	73 (11.0)	590 (89.0)	
Age group, years				0.420
19–29	180 (27.1)	14 (19.2)	166 (28.1)	
30–39	143 (21.6)	15 (20.5)	128 (21.7)	
40–49	168 (25.3)	24 (32.9)	144 (24.4)	
50–59	100 (15.1)	12 (16.4)	88 (14.9)	
60 +	72 (10.9)	8 (11.0)	64 (10.8)	
Occupation				0.834
Non-manual workers	341 (51.4)	36 (49.3)	305 (51.7)	
Sales/Service workers	100 (15.1)	11 (15.1)	89 (15.1)	
Manual workers	28 (4.2)	2 (2.7)	26 (4.4)	
Others	194 (29.3)	24 (32.9)	170 (28.8)	
Marital status				0.002
Single	297 (44.8)	20 (27.4)	277 (46.9)	
Married	366 (55.2)	53 (72.6)	313 (53.1)	
Living arrangement				0.239
Not living alone	580 (87.5)	67 (91.8)	513 (86.9)	
Living alone	83 (12.5)	6 (8.2)	77 (13.1)	
History of pregnancy and childbirth				<0.001
No history	299 (45.1)	20 (27.4)	279 (47.3)	
Pregnancy only	39 (5.9)	2 (2.7)	37 (6.3)	
Pregnancy and childbirth	325 (49.0)	51 (69.9)	274 (46.4)	
Depression				0.008
No	351 (52.9)	28 (38.4)	323 (54.7)	
Yes	312 (47.1)	45 (61.6)	267 (45.3)	
Perceived Stress				0.102
Low	59 (8.9)	5 (6.8)	54 (9.2)	
Moderate	246 (37.1)	20 (27.4)	226 (38.3)	
High	358 (54.0)	48 (65.8)	310 (52.5)	
Age of smoking initiation, years *	25.7 (± 8.8)	26.5 (± 8.5)	25.6 (± 8.8)	0.375
Current CC use				0.390
Non-user	121 (18.3)	16 (21.9)	105 (17.8)	
User	542 (81.7)	57 (78.1)	485 (82.2)	
Current EC use				0.172
Non-user	514 (77.5)	52 (71.2)	462 (78.3)	
User	149 (22.5)	21 (28.8)	128 (21.7)	
Current HTP use				0.003
Non-user	432 (65.2%)	36 (49.3%)	396 (67.1%)	
User	231 (34.8%)	37 (50.7%)	194 (32.9%)	
Multiple tobacco product use				0.001
Current single user	388 (58.5%)	30 (41.1%)	358 (60.7%)	
Current multiple user	275 (41.5%)	43 (58.9%)	232 (39.3%)	
Parent(s) smoking status				0.019
No	477 (71.9)	44 (60.3)	433 (73.4)	
Yes	186 (28.1)	29 (39.7)	157 (26.6)	
Partner smoking status				0.075
No	391 (59.0)	36 (49.3)	355 (60.2)	
Yes	272 (41.0)	37 (50.7)	235 (39.8)	
Friend(s) smoking status				0.500
No	297 (44.8)	30 (41.1)	267 (45.3)	
Yes	366 (55.2)	43 (58.9)	323 (54.7)	
Received advice to quit				0.030
No	103 (15.5)	5 (6.8)	98 (16.6)	
Yes	560 (84.5)	68 (93.2)	492 (83.4)	

Abbreviation: CC, conventional cigarette; EC, electronic cigarette; HTP, heated tobacco product. * Results are shown as mean (± standard deviation). ^†^ Significance at *p* < 0.05.

**Table 2 ijerph-18-05313-t002:** Simple and multiple regression analyses of factors associated with the utilization of national tobacco cessation services (*n* = 663).

Variables	Cases (*n* = 73)	Controls (*n* = 590)	Simple		Model 1 *		Model 2	
			OR	(95% CI)	aOR	(95% CI)	aOR	(95% CI)
Marital status								
Single	20	277	1.00		1.00		1.00	
Married	53	313	2.35	(1.37, 4.02)	1.05	(0.44, 2.49)	1.04	(0.44, 2.47)
History of pregnancy and childbirth								
No history	20	279	1.00		1.00.		1.00.	
Pregnancy only	2	37	0.75	(0.17, 3.36)	0.71	(0.15, 3.28)	0.71	(0.15, 3.30)
Pregnancy and childbirth	51	274	2.60	(1.51, 4.47)	2.37	(0.94, 6.00)	2.34	(0.92, 5.94)
Depression								
No	28	323	1.00.		1.00.		1.00.	
Yes	45	267	1.94	(1.18, 3.20)	2.02	(1.21, 3.40)	2.03	(1.21, 3.40)
Current HTP use								
Non-user	30	119	1.00.		1.00.		1.00.	
User	32	237	2.10	(1.29, 3.42)	1.52	(0.82, 2.83)	1.57	(0.84, 2.93)
Multiple tobacco product use								
Current single user	33	139	1.00.		1.00.		1.00.	
Current multiple user	29	217	2.21	(1.35, 3.63)	1.53	(0.82, 2.85)	1.50	(0.80, 2.81)
Parent(s) smoking status								
No	25	92	1.00.		1.00.		1.00.	
Yes	27	264	1.82	(1.10, 3.01)	1.85	(1.07, 3.18)	1.84	(1.07, 3.17)
Received advice to quit								
No	57	284	1.00.		1.00.		1.00.	
Yes	5	72	2.71	(1.06, 6.89)	2.09	(0.80, 5.46)	2.10	(0.80, 5.51)
Interaction terms								
Current HTPs use × age							0.98	(0.92, 1.04)
Multiple tobacco product use × age							1.02	(0.96, 1.08)
Nagelkerke R^2^					0.129		0.130	

Abbreviation: OR, odds ratio; 95% CI, 95% confidence interval; HTP, heated tobacco product. * Adjusted for age (linear and quadratic terms), marital status, history of pregnancy and childbirth, depression, current HTP use, multiple tobacco product use, parent(s) smoking status and receipt of advice to quit.

**Table 3 ijerph-18-05313-t003:** Multinomial logistic regression analysis of factors associated with the number of utilized service type (*n* = 663).

Variables	Never Used (*n* = 590)	Single Type (*n* = 54)	Multiple Type (*n* = 19)	Single Type vs. Never Used *		Multiple Type vs. Never Used *	
				OR	(95% CI)	OR	(95% CI)
Marital status							
Single	277	16	4	1.00.	1.00.	1.00.	1.00.
Married	313	38	15	1.31	(0.49, 3.51)	0.60	(0.11, 3.24)
History of pregnancy and childbirth							
No history	279	17	3	1.00.	1.00.	1.00.	1.00.
Pregnancy only	37	1	1	0.45	(0.06, 3.68)	1.95	(0.19, 20.46)
Pregnancy and childbirth	274	36	15	2.22	(0.80, 6.16)	3.25	(0.46, 23.15)
Depression							
No	351	21	7	1.00.	1.00.	1.00.	1.00.
Yes	312	33	12	2.06	(1.15, 3.69)	2.30	(0.87, 6.09)
Current HTP use							
Non-user	396	27	9	1.00.	1.00.	1.00.	1.00.
User	194	27	10	1.58	(0.78, 3.18)	1.24	(0.41, 3.79)
Multiple tobacco product use							
Current single user	358	30	6	1.00.	1.00.	1.00.	1.00.
Current multiple user	232	24	13	1.41	(0.70, 2.84)	2.83	(0.88, 9.09)
Partner smoking status							
No	355	31	5	1.00.	1.00.	1.00.	1.00.
Yes	235	23	14	0.87	(0.47, 1.60)	3.24	(1.06, 9.87)
Nagelkerke R^2^				0.121			

Abbreviation: OR, odds ratio; 95% CI, 95% confidence interval; HTP, heated tobacco product. * Adjusted for age (linear and quadratic terms), history of pregnancy and childbirth, depression, current HTP use and multiple tobacco product use.

**Table 4 ijerph-18-05313-t004:** The survey participants’ motivation ruler for smoking cessation and concern about weight gain (*n* = 579).

Variables	National Tobacco Cessation Services			*p* Value *	Age Category				
	Total	Ever Used	Never Used		19–29	30–39	40–49	50–59	60+
	579 (100)	56 (14.7)	325 (85.3)		158 (27.3)	122 (21.1)	150 (25.9)	85 (14.7)	64 (11.1)
Importance ruler †	7.4 (± 2.2)	8.2 (± 1.9)	7.2 (±2.2)	0.003	6.7 (± 2.5) ^a^	7.2 (± 2.1) ^ab^	7.1 (± 2.2) ^ab^	7.7 (± 2.0) ^b^	7.3 (± 2.0) ^ab^
Confidence ruler †	6.2 (± 2.2)	6.6 (± 1.9)	6.2 (± 2.2)	0.165	6.1 (± 2.5)	5.9 (± 2.0)	5.7 (± 2.2)	6.6 (± 1.9)	5.7 (± 2.1)
Readiness ruler †	6.5 (± 2.1)	6.8 (± 2.3	6.5 (± 2.1)	0.203	6.2 (± 2.3)	6.2 (± 2.0)	6.0 (± 2.0)	6.7 (± 2.0)	5.9 (±2.1)
Post-cessation weight gain concerns †	6.1 (± 2.8)	7.1 (± 2.5)	5.9 (± 2.9)	0.005	4.9 (± 3.0) ^a^	6.2 (± 2.8) ^b^	6.6 (± 2.5) ^b^	6.3 (± 2.3) ^b^	6.2 (± 2.6) ^b^

Results are shown as mean (± standard deviation) or number (%). * Significance at *p* < 0.05. ^†^ Motivation rulers (importance, confidence, readiness) and weight gain concerns were assessed by an 11-point Likert scale (0 = not at all concerned to 10 = very concerned). ^a,b^ Values not sharing the same uppercase letter in a row indicate significant statistical differences among groups based on Scheffé’s test.

**Table 5 ijerph-18-05313-t005:** The survey participants’ need for gender-specific smoking cessation services (*n* = 663).

Variables	National Tobacco Cessation Services			*p* Value *	Age Category				
	Total	Ever Used	Never Used		19–29	30–39	40–49	50–59	60+
	663 (100)	73 (11.0)	590 (89.0)		180 (27.1)	143 (21.6)	168 (25.3)	100 (15.1)	72 (10.9)
Need for gender-specific smoking cessation services ^†^	3.8 (± 0.8)	4.0 (± 0.7)	3.6 (± 0.9)	< 0.001	3.5 (± 0.9) ^a^	3.6 (± 0.9) ^ab^	3.9 (± 0.7) ^b^	3.8 (± 0.8) ^ab^	3.7 (± 0.9) ^ab^

Results are shown as mean (± standard deviation) or number (%). * Significance at *p* < 0.05. ^†^ Needs were assessed by a 5-point Likert scale (1 = strongly disagree to 5 = strongly agree). ^a,b^ Values not sharing the same uppercase letter in a row indicate significant statistical differences among groups based on Scheffé’s test.

## Data Availability

The data presented in this study are available on request from the corresponding author. The data are not publicly available to protect confidentiality of the research participants.

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
