# Peer review of "The Utilization of National Tobacco Cessation Services among Female Smokers and the Need for a Gender-Responsive Approach"

_ijerph, 2021, doi:10.3390/ijerph18105313_

Round 1

Reviewer 1 Report

  1. This research presents many statistics on female smokers’ use of Korea’s cessation services. The analysis uses data from a novel sample and focuses on the need for gender-differentiated stop-smoking services. The authors find several differences among different groups of women (age groups, reproductive history, etc.) in usage of services and attitudes. From these results, the authors conclude that there are “several gendered factors related to the utilization of national health services” and that the results can help design better services for women.

    1. I find that the basic premise of the paper needs a better explanation. I would have thought that if the goal was to explore “gendered factors related to the utilization of national health services” that one would study men and women and show how their factors differed. Concern about weight gain is not limited to women. Co-use of ENDS and HTP is not specific to women. Pregnancy and childbirth are only partially specific to women, since having a new child in the home also motivates some men to quit. “Gendered factors” is not a synonym for “we studied women only”.
    2. The analysts are not specific on what their results are meant to represent. The tacit assumption is that the results are reflective of the subpopulation of female smokers, but that is not correct (or, at least, they haven’t shown that it is correct). The sample is apparently a random sample from an online panel collected who-knows-how, and the characteristics of the sample are never compared with those from high-quality surveys such as KNHANES. There is no mention of weights or weighting in the analysis, and so it seems that all the statistics are unweighted. So, the results really reflect only the sample, not any specified population. This should be clarified, and it wouldn’t hurt to compare the sample with known marginals from (weighted) KNHANES data, if they want to claim the sample is truly representative. At a minimum, the authors need to clarify that all of their inference is for the sample, not the population. They have not used design-based methods to compute standard errors that would be needed for the latter.
    3. The authors conclude that "Given the crucial initial findings of this study, further research with sufficient power to assess this association is needed". First, I do not think they have made the case for why their results are “crucial”. Crucial for what, exactly? And what is “this association”? This sentence needs to be made more specific, since it seems to be meant to reflect the whole point of the research.
    4. The paper would benefit from some discussion on gender-nuanced services. At the start of the paper it seems that they would be a good approach and necessary. Later in the paper it becomes clear that there are potentially unhelpful ways to gender-differentiate cessation services (i.e., overemphasizing harm to unborn children vs. caring about the woman in her own right). So clearly the call to action cannot just be to “do things differently” for female smokers.
    5. Finally, and perhaps related to point #1 above, the usage rates for ENDS and HTP are very high compared with the latest KNHANES data, I believe (at least the latest rates I saw). Why the discrepancy? Did usage rates really climb so much among women in such a short time?

    Other comments:

    1. "female smokers tend to be reluctant to reveal their smoking status": Yes, this is known for Korean smokers. So, this should be discussed when the earlier statistics on female smoking rates are presented. Those rates cannot be believed.
    2. Some of the Korean sources in the reference list appear not to be in English, but this isn’t noted in the citation. E.g. ref. 2, which I couldn’t find. Authors should consult a style guide for proper citation of non-English sources (or the editors can provide guidance).
    3. Reference 20 needs the issue number, since pagination begin at 1 in each issue.
    4. "Hopeless" on p.3 should be “hopelessness”.
    5. Since the authors discuss cessation in Korea, it is probably worth citing a new omnibus report on this topic, since it has a comprehensive review of all aspects of the literature: see https://papers.ssrn.com/sol3/papers.cfm?abstract_id=3773245.
    6. "Notably, in our data, participants in their 20s were less in favor of female-focused smoking cessation services". But the table shows that 1) the differences are small, and 2) that age group isn't significantly different from any but the highest group. So is this really “notable”? And even if it is, why?
    7. Can the authors clarify the statistical work underlying Table 3? I think that there is a separate logistic regression for each categorical variable (with and without age as additional regressors), but the reader shouldn’t have to guess.

Author Response

We would like to thank the reviewer for the thoughtful comments, which we found helpful in improving the quality of the manuscript. We have provided a point-by-point format responses to all comments by the reviewer. Changes made to the manuscript are indicated in highlighted texts in the revised manuscript and "Track Changes" function in Microsoft Word were used. Please refer to the attached file.

Reviewer 2 Report

This article focuses on understanding how smoking cessation programs impact female smokers specifically.  Using a survey of smokers in Korea, it looks at the factors that increase the propensity by which female smokers will utilize a cessation program.  This novel study finds that several factors are related to the propensity to use a smoking cessation program, namely pregnancy status, depression, user of multiple tobacco products, parent smoking, and whether the respondent received advice to quit.

For me, the biggest shortcoming was the intentional decision to exclude men from the study.  To identify differences between men and women, it would be advantageous to have also included men in the study.  Then, a simple indicator variable could be used to distinguish the role of various factors on smoking cessation.  Interaction terms with gender could also be supported.  In any event, short of conducting another survey with this sample (complicated because of the time differences)- I would at least like to know why that decision was made.

Of secondary concern, I would like to know more about the final sample relative to those that responded and the general population.  For example, the authors note that 290 of the 708 respondents were excluded due to missing values.  What was the nature of missingness?  Were missing values associated with key factors?  Also, is the sample representative of the population at large in terms of key demographics such that the results could generalize to the population of interest?

In terms of the modelling, do the tables represent a full multivariate analysis or a set of bi-variate analyses?  I'm hoping the former, so that other factors are controlled and accounted for.  Also, could the respondents report the overall model fit (adjusted R2) and N for the models?  The authors might consider, even if not for this paper, using an IRT measure for their dependent variable.  Maybe respondents used more than one cessation program- and the authors could look at the intensity of the usage of cessation rather than a dichotomous dependent variable. The authors might consider doing the same for their different "Rulers" if they are aggregates.

It seems to me that at least in some countries, I don't know if this is true in Korea, that smokers might also be associated with drinkers.  I wondered why this might not be controlled for. 

A minor question- is age a continuous variable?  If so, I wondered if the authors considered using quadratics.  Maybe likelihood of cessation changes as a function of age but nonlinearly. 

Finally, I wondered if there were any interaction terms that might be important (example: Age*multiple tobacco user)?

I think this is an important question and I commend the authors for taking it up and for presenting the results so clearly in the tables. 

Author Response

(The authors gave the same response as above.)

Reviewer 3 Report

This study examines factors associated with the utilization of national smoking cessation services in Korea and determines whether there was a gender gap. This is a pertinent public health topic with potential important policy implication, especially in Korea, where the gender gap in smoking is different from the Western countries. 

  1. Line 95: 708 Korean female ever-smokers over the age of 19 were randomly sampled. A nationwide web survey was conducted: please explain how a random sample was obtained using a web survey.
  2. 290 out of 708 participants were excluded based on missing data. As a results, even if the original 708 participants were recruited randomly, the final sample cannot be a random sample.
  3. Participants who selected any of the smoking cessation services were categorized as users: this definition may not truly capture degree of utilization as there may be an intensity problem: females may have lower number of users but may at the same time have higher intensity of use. Can this also be analyzed?
  4. Regression analysis: it is unclear why certain variables were included as confounders, for example, stress level and depression. These factors are likely to be a consequence of service use, instead a confounder of service use. Current smoker: for sure this cannot happen before using the program as only smokers were included. This making the regression model flawed. I think the author needs to come up with a conceptual framework before analyzing the variables and explain clearly their hypothesis.
  5. Table 3: Odds ratios across models should never be compared in terms of magnitude. If the authors wish to compare models, marginal effects should be presented instead of odds ratios.
  6. Line 54: “The utilization of smoking cessation treatments is regarded as a cost-effective way to 54 reduce smoking-induced morbidity and mortality”: did the author really mean “utilization” as an effective way?

Author Response

(The authors gave the same response as above.)

Round 2

Reviewer 2 Report

I commend the authors on this nice work and for taking the reviews seriously.  Your thoughtfulness in response and greater clarity helped improve the manuscript.  I now better understand the data generating process and the limitations of those data.  Thank you for also taking the time to test out the age effects, which seemed important to me.  

Reviewer 3 Report

The authors have responded to my concerns. I think the authors did the best they could given the data they have.